# Assessing Mental Health Outcomes in Quarantine Centres: A Cross-Sectional Study during COVID-19 in Malaysia

**DOI:** 10.3390/healthcare11162339

**Published:** 2023-08-18

**Authors:** Nadia Mohamad, Rohaida Ismail, Mohd Faiz Ibrahim, Imanul Hassan Abdul Shukor, Mohd Zulfinainie Mohamad, Muhammad Farhan Mahmud, Siti Sara Yaacob

**Affiliations:** 1Environmental Health Research Centre, Institute for Medical Research, National Institutes of Health, Shah Alam 40170, Selangor, Malaysia; 2Non-Communicable Disease Control Sector, Selangor State Health Department, Ministry of Health Malaysia, Shah Alam 40100, Selangor, Malaysia; 3Department of Public Health Medicine, Faculty of Medicine, Universiti Teknologi MARA, Sungai Buloh 47000, Selangor, Malaysia

**Keywords:** mental health outcomes, anxiety, quarantine centres, frontline workers, person under surveillance, COVID-19

## Abstract

During the COVID-19 pandemic, persons under surveillance (PUS) were isolated in quarantine centres instead of at home. However, there is limited knowledge regarding the mental health issues experienced by these persons. This study aimed to assess mental health outcomes and associated factors among PUS and frontline workers at quarantine centres. This study conducted an analysis of secondary data from a cross-sectional survey carried out by the Mental Health and Psychosocial Support Services (MHPSS). The MHPSS employed the Depression, Anxiety, and Stress Scale (DASS-21) to evaluate mental health outcomes across 49 quarantine centres in Malaysia. The study included a total of 4577 respondents. The prevalence of stress, anxiety, and depression was found to be 0.9%, 11.4%, and 10.2%, respectively. Frontline workers and being part of the younger age group were found to be associated with depression, anxiety, and stress. Other factors associated with mental health issues were being female, staying at an institution-type centre, and a longer duration of the stay or work at the centre. In conclusion, assessing the mental health status and its associated factors among quarantine centre occupants is crucial for developing future strategies to safeguard their mental well-being.

## 1. Introduction

Coronavirus disease of 2019 (COVID-19), a highly infectious disease that originated in Wuhan, China in 2019, was declared a global pandemic by the World Health Organization in March of 2020 [1]. Lockdown measures were implemented worldwide to limit human interactions, but the strategy proved unsustainable in the long run. The COVID-19 pandemic had a devastating global impact on people’s lives, leading to loss of jobs, closure of shops, forced separation from loved ones, and adverse mental health problems [2]. The situation worsened with the emergence of the Delta variant, resulting in a surge of cases and deaths worldwide [3]. The overwhelmed healthcare system likely led to an underestimation of COVID-19 deaths, and the uncertainty of the future led to an increase in suicide rates, especially among youth [4].

Providing mental health support to vulnerable groups, particularly in healthcare facilities and quarantine centres, is crucial, especially during the challenging times of the COVID-19 outbreak. In an effort to comply with the public health response to the pandemic and to mitigate COVID-19 transmission, significant adjustments and adaptations were made to deliver mental health services. In Malaysia, one of the initiatives instituted to tackle this problem was the implementation of the Mental Health and Psychosocial Support Services (MHPSS) that access a wide range of technologies utilising telecommunications and internet resources [5]. This system was implemented across all centres for quarantined individuals. The MHPSS team was formed at each quarantine station to provide psychological support services around the clock. The target groups were the COVID-19 low risk patients, person under investigation (PUI), person under surveillance (PUS), healthcare workers (HCW), and other responders from agencies that were directly involved in managing quarantine stations [6].

Early in 2020, the global spread of COVID-19 prompted the Malaysian government to take unexpected containment measures. The Movement Control Order (MCO) was issued, proclaiming an absolute ban on leaving one’s house for any reason other than for required employment or to obtain basic necessities [7]. The MCO interrupted Malaysians’ normal lives, forcing them to retreat from society and isolate themselves. In accordance with precautionary measures, PUS were quarantined for 14 days in hotels or institutional facilities. However, due to the lengthy quarantine period, the environment in quarantine centres may not have been conducive to meeting the ongoing physical, mental, and medical needs of the individual, potentially causing psychological stress. Studies show that mental health problems and associated issues varied significantly among different groups, including the general public, individuals with COVID-19, and healthcare providers [8,9]. Thus, the MHPSS played a very important role in addressing the mental health aspects of COVID-19, especially among persons under quarantine. Nevertheless, limited data was available regarding the psychological conditions in quarantine centres during the COVID-19 pandemic. This study aimed to assess the mental health-related issues among PUS and frontline workers in quarantine centres during the second wave of the COVID-19 pandemic in Malaysia. In addition, this study attempted to identify areas for service improvement at quarantine centres, from a PUS perspective.

## 2. Materials and Methods

This cross-sectional study was conducted during the second wave of the COVID-19 pandemic in Selangor. The quarantine centres were established in early March 2020 to accommodate PUS, defined as persons at risk of being infected with COVID-19 if they: (1) have traveled from overseas to Malaysia and (2) had a history of contact with confirmed COVID-19 cases, but have not yet tested positive for COVID-19. Most PUS were quarantined for a period of 14 days, but a few were discharged later, as their COVID-19 symptoms persisted beyond the stipulated period.

### 2.1. Study Area and Study Population

Selangor, a highly populated state in Malaysia, is home to more than 6 million people. It is also known as the most developed state in Malaysia, with most of the labor force concentrated in this region. The main entry point to Malaysia, the Kuala Lumpur International Airport (KLIA), is located in Selangor. During the containment period, one of the national policies was to quarantine all the travelers entering the country in quarantine centres until the incubation period was over before releasing them into society. Due to its location and its associated risks, Selangor had more quarantine centres compared to other states in Malaysia. Selangor also reported the highest number of positive cases of COVID-19. Although the present study only considered the state of Selangor, some respondents in this study came from other states, but were quarantined in Selangor. Mental health screening and psychological support were offered by the MHPSS team to all PUS and frontline workers in quarantine centres. The inclusion criteria were (1) PUS who were quarantined or frontliners who worked at quarantine centres in Selangor during the first six months of the pandemic outbreak, from March to August 2020, and (2) adults, aged 18 years old and above. The exclusion criteria were those with existing psychiatric disorders. Throughout the study period, a total of 1354 frontline workers attended to 10,655 PUS in 49 quarantine centres, of which 5454 subjects provided mental health screening data. However, 877 responses were excluded (774 = duplicate data; 103 = under 18 years), leaving the final number of 4577 responses to be included in this study (Figure 1).

### 2.2. Data Collection and Materials

The anonymized mental health screening data from March to August 2020 were retrieved from the MHPSS team, and the data collection process was facilitated by the Non-Communicable Unit, Selangor State Health Department. The recruitment process was accomplished through trained medical personnel at the quarantine centers by distributing a questionnaire link using posters and social media. The self-administered questionnaire was delivered to the participants using online forms. To maintain anonymity, the MHPSS team converted personal details into anonymous identification codes prior to transferring the data to researchers in a specifically designed format.

Data extracted from the MHPSS were divided into three parts: (1) sociodemographic characteristics; (2) depression, anxiety, and stress; and (3) services at quarantine centres and personal issues that require psychological assistance. Firstly, the independent variables were gathered, based on sociodemographic characteristics (gender, age, ethnicity, nationality, and occupational status), including the quarantine details (status as frontline workers or PUS, duration of stay, and type of centre).

Secondly, dependent variables (mental health outcomes) were retrieved based on the MHPSS psychological assessment using the Depression, Anxiety, and Stress Scale (DASS-21) [6]. The validity and reliability of DASS-21 indicated its use as a mental health screening tool during the COVID-19 pandemic for communities, quarantined individuals, and frontline workers [10,11,12,13]. The DASS-21 has been translated into various languages, including Malay, and this version was used in diverse samples, either in the general population or in clinical settings [14,15,16,17]. There are seven items on each subscale of the DASS-21 that assess symptoms for (1) depression: dysphoria, hopelessness, self-worthlessness, and lack of interest; (2) anxiety: somatic symptoms, situational anxiety, and subjective experience of an anxious affect; and (3) stress: persistent arousal and tension consisting of symptoms such as difficulty in relaxing, agitation, irritability, and impatience [18]. There were four responses provided for each item: 0 (never), 1 (sometimes), 2 (a lot of the time), and 3 (most or all of the time). All items were then added together as a total score, and the responses were divided into five categories: none, mild, moderate, severe, and extremely severe.

The last part of the MHPSS data focused on the services at the quarantine centres (facilities and food improvement) and personal issues that required psychological assistance (financial, family, career, health, childcare, and others). The respondents were requested to choose the aspects of the quarantine centre that they believed could be improved.

### 2.3. Data Analysis

All retrieved data were transferred to Microsoft Excel, and further analysed using the Statistical Package for the Social Sciences (SPSS) version 21 (IBM, Armonk, NY, USA). The descriptive data regarding the sociodemographic characteristics and mental health outcomes were tabulated as frequencies (n) and percentages (%) for category data, while continuous data were expressed in means and standard deviations (±SD). All statistical tests were two-sided, and a *p* value less than 0.05 (*p* < 0.05) was considered significant. Univariate and multivariate analyses were performed using binary logistic regression to explore the independent effects for the different categorical variables (sociodemographic and quarantine characteristics) on dependent variables (stress, anxiety, and depression), and the covariates were determined in the best fit model. Independent variables chosen as references were coded as 1, and *p* < 0.05 was considered as statistically significant. Multicollinearity terms were checked, and the Hosmer–Lemeshow test and classification table were applied to check for model fitness.

### 2.4. Ethical Issues

This study primarily involved secondary data analysis from the DASS-21 and feedback forms received from the participants during their stay in the quarantine centres. Informed consent was obtained when participants responded to the survey (DASS and feedback form) conducted through the MPHSS programme. To ensure privacy and confidentiality, the authors obtained anonymised data from the MPHSS team for further secondary data analysis.

## 3. Results

This survey was completed by 5454 respondents at the Selangor quarantine centres. Of the 5454 respondents, a total of 4577 were included in this study, where 4122 (90.1%) were PUS, while another 455 (9.9%) were frontline workers. The majority of the respondents were male (57.0%), stayed or worked at 4-star hotels (49.3%), were of Malaysian nationality (92.6%) and Malay ethnicity (42.4%), and were working adults (54%). The mean age was 33 ± 12 years, and the mean number of days in the quarantine centre was 6 ± 6 days. Table 1 describes the characteristics of the respondents from the quarantine centres.

Table 2 summarizes the level of stress, anxiety, and depression among PUS and frontline workers. Overall, PUS exhibit a more normal level of mental health compared to frontline workers, whereas moderate levels of stress, anxiety, and depression were more common in frontline workers than in PUS. There is a significant difference between these two groups in regards to the levels of anxiety and depression (*p* < 0.05).

Simple and multiple logistic regression analyses were conducted to determine factors associated with stress, anxiety, and depression in the quarantine centres, and these are tabulated in Table 3. The final model demonstrated the absence of multicollinearity. The model was fitted based on the Hosmer-Lemeshow tests, which were not significant for all the models (*p* > 0.05). Additionally, the classification table showed an overall correctly classified percentage of more than 70% for all models. According to the analysis, being 18 to 30 years old and being frontline workers were the factors indicated as statistically significant for all the outcomes, namely stress, anxiety, and depression, with AOR = 2.02, 95% CI = 1.2–3.5, AOR = 1.60, 95% CI = 1.1–2.4, and AOR = 1.97, 95% CI = 1.3–2.0, respectively, when compared to the results for responders more than 50 years old. The multivariate analysis for frontline workers indicated that AOR = 1.94, 95% CI = 1.2–3.1 for stress, and AOR = 1.99, 95% CI = 1.4–2.8 for anxiety, and that AOR = 1.90, 95% CI = 1.3–2.7 for depression, compared to the reference group (PUS). Being female was found to be more associated with stress (AOR = 1.51, 95% CI = 1.2–2.0) and anxiety (AOR = 1.34, 95% CI = 1.1–1.6) than being male. The types of quarantine centres and the days of quarantine were associated with anxiety, after adjusting the odds. The respondents were found to have a 1.7 times greater probability for anxiety when the quarantine centre was an institution rather than a hotel, with the odds of anxiety being 1.3 times greater when the stay was longer than 7 days. In contrast, respondents who were employed exhibited a protective effect against anxiety (AOR = 0.6, 95% CI = 0.44–0.88) and depression (AOR = 0.6, 95% CI = 0.41–0.85) compared to respondents who were not working or retired.

From the PUS who answered the survey, a total of 372 responded to survey regarding whether or not they required psychological assistance. Figure 2 shows the types of problems faced by the PUS requiring psychological assistance.

In addition, 237 PUS provided suggestions regarding the services or facilities in the centres, where 111 PUS suggested facility upgrades, and 129 suggested food improvement. Areas of improvement for food services at the quarantine centres are summarized in Figure 3a. The top suggestion was to improve the menu, as PUS exhibited different food requirements, based on their diverse ethnicities, cultures, religions, and beliefs. Next was in regards to the food’s quality and taste, as well as suggestions that the facility provide a more healthy and balanced diet, followed by recommendations that the facility provide snacks or desserts, in addition to the main meal. Sometimes, PUS opted for more variety, including the option of procuring outside food, either homecooked by their families, or purchased from restaurants, and 17 of them suggest leniency in allowing for the delivery of outside food. All PUS were given the same amount of food, three times a day. However, some may have had different patterns of eating, in term of quantity and frequency. This issue was vocalized by 14 PUS, who asked the centres to increase the portion or frequency of food provided. Lastly, 10 PUS requested more options for drinking water.

Figure 3b summarizes the suggestions to improve the centre’s services or facilities. The most frequent suggestion was to allow exercise or outdoor activities, followed by the provision of more entertainment. Some quarantine rooms did not even have a television, while some respondents who did have access to television requested more channels for their screen time. Twenty PUS requested a room with better ventilation, with access to fresh air and sunlight. This issue arose due to the structure of the building, with fixed windows that do not open, or windowless rooms. Another suggestion was to provide entertainment for the PUS to occupy their time during quarantine. Moreover, being confined in a room, especially without fresh air, was difficult for smokers, and 14 PUS suggested that a smoking area be provided for them. Twenty PUS suggested improving the quality of the housekeeping or the information delivery systems. Whether quarantined in an institution or in a hotel, their status as PUS meant that no room services were provided. However, the centres could provide cleaning apparatus, such as a sweeper, for them to clean their rooms by themselves. Other suggestions included being allowed to quarantine with their friends or family to reduce boredom.

## 4. Discussion

To the best of the authors’ knowledge, this is the first study to evaluate the stressors imposed on PUS and frontline workers while staying or working in quarantine centres in Malaysia during the COVID-19 pandemic. The data from 4577 selected quarantine centres in Selangor were analysed, revealing that 5.9% of the respondents experienced stress, 11.4% felt anxiety, and 10.2% suffered depression. The prevalence of stress, anxiety, and depression noted in this study is much lower than that documented in a study conducted in Saudi Arabia, in which the prevalence of stress, anxiety, and depression was 41.1%, 31.9%, and 31.4%, respectively [19]. Another study in Bangladesh also showed a higher prevalence of stress, anxiety, and depression compared to the that seen in the present study. In that study, 28.5% of the respondents reported experiencing stress, 33.3% reported feeling anxiety, and 46.92% reported suffering depression [20]. Interestingly, the current study also discovered that stress, anxiety, and depression were more prevalent in PUS. This is most likely due to the fact that the healthcare workers were better prepared for and knowledgeable about COVID-19 than were the non-healthcare workers, especially in light of prior coronavirus epidemics in Malaysia, such as SARS and MERS [21].

Mental health issues are typically more prevalent among females than males [22,23,24]. Consistent with other studies, the current findings also indicate that during COVID-19, females experience a higher level of psychological distress than males [25,26,27,28,29]. Females are more likely than males to be concerned about and fearful of COVID-19 [29], two cognitive characteristics that are strongly associated with anxiety and depressive disorders [30]. Additionally, females are more likely to develop psychological symptoms following a traumatic or stressful experience [31]. This suggests that the psychological stress of quarantine may have contributed to the increased risk of mental illness among females. According to a study conducted in Tangshan, China, using the self-rating anxiety scale (SAS), females experienced greater psychological distress than males after seven days of isolation [32]. This psychological gender gap may be influenced by genetic or hormonal factors, as well as structural gender inequality at the societal level [33].

This study found that young adults (18 to 30 years old) are more vulnerable to psychological conditions in quarantine during the COVID-19 pandemic. When compared to older people, young adults are significantly more likely to experience anxiety, stress, and depression. These findings are consistent with previous research from Austria, the United States, the Philippines, and Brazil [34,35,36,37]. Various factors may contribute to these findings, including more uncertain working conditions, which can lead to serious financial problems for young people [38]. Although this is not clear from the survey, adults aged 18 to 30 are typically the ones caring for children and elderly parents, and they are likely concerned about their safety [39]. A previous study has found that, while older adults are more likely to contract COVID-19, they are also more likely to adapt well to the “new norm,” including isolation and other official directives, due to their fear of transmission and death [40]. Another possibility is that older adults place a higher value on their cultural and religious beliefs, which were associated with lower stress levels among believers during the pandemic lockdown [40,41].

Most PUS responded that they sought psychological help due to financial and health issues. However, due to the nature of the survey, a correlation between the reason for seeking psychological assistance and stress, anxiety, or depression could not be established. However, previous research has found that fears of infection and financial loss due to quarantine are risk factors for psychological disorders [42]. The types of financial pressures varied by country and did not affect all individuals equally. For example, a study conducted in Bangladesh revealed that financial stressors were due to decreased household income, food scarcity, and the possibility of unemployment [43]. Nevertheless, research conducted in a high-income nation, the Netherlands, indicated a correlation between decreased savings, increased debt, and increasing financial stress [44]. The financial stress burden was more significant among low-skilled and self-employed individuals because low-skilled workers are less productive at home and suffer greater earning loss when working from home [45].

Most of the respondents were isolated in hotels during their quarantine period. However, despite having more free time, they were unable to utilise it as intended, due to isolation and restrictions. This confinement in a small space with no outside access can be detrimental to one’s mental health. According to a qualitative study conducted in Australia, New Zealand, and Fiji, guests at quarantine hotels experience a range of unstable feelings and moods during their stay, ranging from uncertainty and worry, to loneliness and boredom, to despair and sadness [46]. Although the present study did not measure boredom among the respondents, recommendations provided by them, such as allowing outdoor activities and providing entertainment, suggest that those in quarantine are feeling bored in the centres. In addition, the findings of this study are consistent with those of previous studies showing that dietary restrictions lead to boredom and decreased psychological health [46,47,48]. Meals in the quarantine hotels were provided by government-contracted catering companies. Respondents described these meals as lacking in variety, flavor, and healthy meal options. Initially, hotel guests were also prohibited from receiving packages from friends and family, as well as outside food deliveries. The unexpected dietary changes and restrictions imposed on the quarantined individuals may have impaired their coping abilities and mental health [46,49,50]. Furthermore, another unsatisfactory feature of quarantine hotels was the accommodations, often exhibited a lack of natural lighting, fresh air, room cleanliness, and provisions for proper hygiene. Similar variables have also been documented by an Australian study, which ranked operable windows, ventilation, and natural lighting as the top three sources of well-being in quarantine hotels [51]. Nevertheless, pandemics are extraordinary historical events that profoundly alter mental health services and their delivery [52]. An investigation of the availability and effectiveness of digital mental health support under quarantine conditions is needed.

## 5. Recommendation for Policy Makers

Policy makers should prioritise the mental health of those in quarantine during the public health response to COVID-19. Evidence suggests that the negative psychological effects of quarantine can persist for months or even years [53,54,55]. To mitigate these effects, quarantine facilities must be conducive to promoting mental health and offer regular exercise, connections with loved ones, access to work or study online, outdoor access, healthy food, and mental health support. Governments should plan for the availability of these items in quarantine facilities in the event of future epidemics or pandemics.

## 6. Limitations

There are several limitations of this study that must be acknowledged. Firstly, this research examines data gathered from individuals who were quarantined in centres in Selangor, Malaysia. This study is limited to the Malaysia’s quarantine system, and survey results may differ, if conducted in other states or countries. Secondly, the study of secondary data was constrained by the availability of such data. The MHPSS data was collected through a cross-sectional approach and due to the lockdown scenario resulting from the COVID-19 outbreak, an online convenience sampling technique was used, which was not based on a random sample selection. As a result, making causal inferences was impossible, and the risk of sample bias should also be considered. Finally, this study relied on self-reported responses about experiences during mandatory quarantine, which may or may not correspond with the clinical diagnosis of a mental health professional.

## 7. Conclusions

Quarantine can cause depression, anxiety, and stress, especially when subjects are quarantined at a quarantine centre and isolated alone in a room, away from their homes. This study identified the group of people who are at greater risk for developing stress, anxiety, and depression. It is essential to develop preventive strategies to promote mental health wellbeing for this population. Moreover, this study also explored the reasons why PUS at the centres required psychological support, and their suggestions could be used to improve the services there. Even though more thorough study regarding these factors is needed in the future, these findings could be used as baseline data regarding how to support the mental health wellbeing of PUS and frontline workers at quarantine centres, as well as on how to improve these services when facing future disasters.

## Figures and Tables

**Figure 1 healthcare-11-02339-f001:**
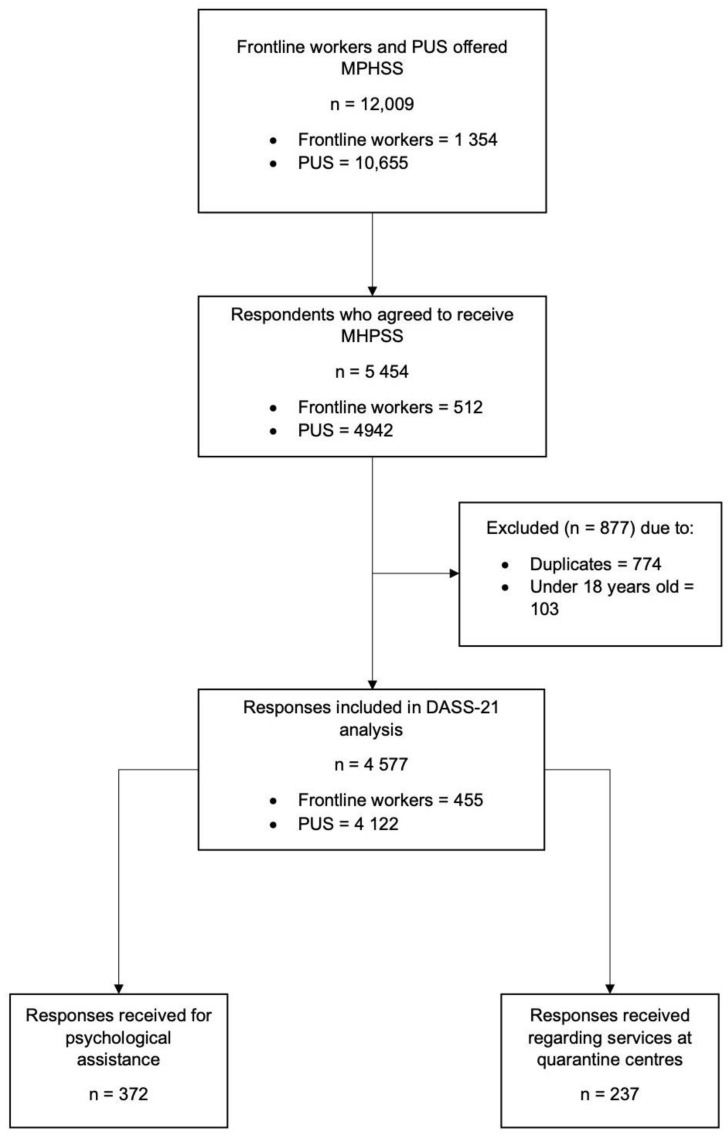
Flowchart of respondents included in the study.

**Figure 2 healthcare-11-02339-f002:**
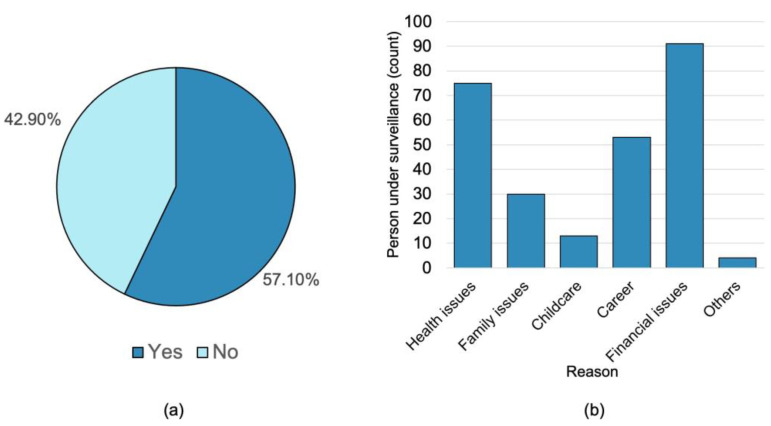
(**a**) Percentage of PUS who required psychological assistance and (**b**) the reason for PUS requiring psychological support.

**Figure 3 healthcare-11-02339-f003:**
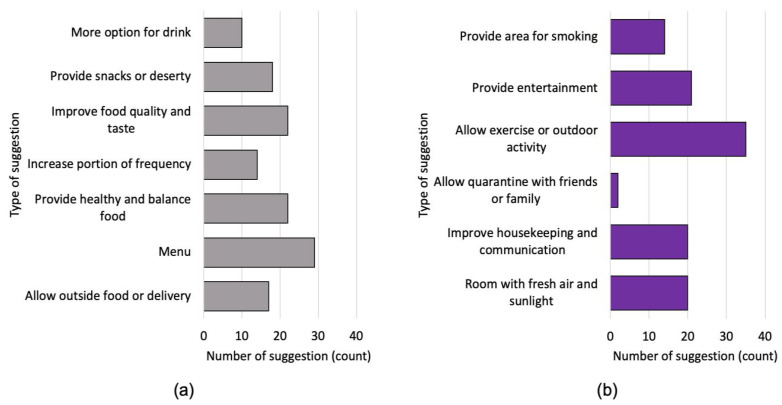
Persons under quarantine suggestions to improve (**a**) the centre’s food; (**b**) the centre’s facilities.

**Table 1 healthcare-11-02339-t001:** Characteristics of person under quarantine and frontline workers at quarantine centres (N = 4577).

Variables	Mean (SD)	n	%
**Gender**			
Male		2609	57.0
Female		1968	43.0
**Age**	33.06 (12.5)		
18-30		2464	53.8
31–40		1032	22.5
41–50		534	11.7
**Days at the quarantine centre**	6.4 (6.087)		
7 days or less		3280	71.7
>7 days		1297	28.3
**Type of Centre**			
5-star hotel		1856	40.6
4-star hotel		2256	49.3
2- or 3-star hotel		141	3.1
Institute		324	7.1

**Table 2 healthcare-11-02339-t002:** Level of stress, anxiety, and depression among the study population.

Variable	Category	None, n (%)	Mild, n (%)	Moderate, n (%)	Severe, n (%)	Extremely Severe, n (%)	*p* Value
Stress	PUS	3886 (94.3)	111 (2.7)	62 (1.5)	41 (1.0)	22 (0.5)	0.101
	Frontline workers	422 (92.7)	14 (3.1)	13 (2.9)	6 (1.3)	0 (0)	
Anxiety	PUS	3671 (89.1)	128 (4.7)	193 (4.7)	66 (1.6)	64 (1.6)	<0.001 *
	Frontline workers	382 (84.0)	12 (2.6)	47 (10.3)	7 (1.5)	7 (1.5)	
Depression	PUS	3714 (90.1)	158 (3.8)	161 (3.9)	42 (1.0)	47 (1.1)	0.020 *
	Frontline workers	398 (87.5)	21 (4.6)	31 (6.8)	3 (0.7)	2 (0.4)	

Note: PUS = person under surveillance; * = *p* value < 0.05.

**Table 3 healthcare-11-02339-t003:** Odds ratio for mental health issues/psychological conditions in the quarantined population.

Variable	Stress			Anxiety			Depression		
	Yes	Univariate	Multivariate	Yes	Univariate	Multivariate	Yes	Univariate	Multivariate
	n (%)	OR (95% CI)	OR (95% CI)	n (%)	OR (95% CI)	OR (95% CI)	n (%)	OR (95% CI)	OR (95% CI)
**Gender**									
Male	117 (4.5)	1	1	247 (9.5)	1	1	234 (9.0)	1	1
Female	152 (7.7)	1.783 (1.390–2.286) **	1.508 (1.160–1.959) **	277 (14.1)	1.566 (1.305–1.880) **	1.335 (1.100–1.620) **	231 (11.7)	1.350 (1.14–1.636) **	1.112 (0.907–1.362)
**Age**									
30 and below	192 (7.8)	2.232 (1.576–3.163) **	2.018 (1.178–3.456) **	352 (14.3)	2.262 (1.576–3.247) **	1.615 (1.078–2.418) *	330 (13.4)	3.008 (2.138–4.232) **	1.973 (1.304–2.985) **
31–40	38 (3.7)	1.295 (0.874–1.920)	1.131 (0.629–2.035)	91 (8.8)	1.064 (0.700–1.620)	1.312 (0.859–2.004)	70 (6.8)	1.798 (1.246–2.593)	1.158 (0.740–1.812)
41–50	19 (3.6)	1.173 (0.745–1.846)	1.138 (0.588–2.202)	43 (8.1)	0.871 (0.527–1.440)	1.320 (0.823–2.120)	30 (5.6)	1.282 (0.841–1.956)	0.992 (0.590–1.668)
Greater than 50	20 (3.7)	1	1	38 (6.9)	1	1	35 (6.4)	1	1
**Centre**									
Hotel	252 (5.9)	1	1	467 (11.0)	1	1	427 (10.0)	1	1
Institution	17 (5.2)	0.879 (0.531–1.456)	0.809 (0.456–1.435)	57 (17.6)	1.731 (1.280–2.340) **	1.699 (1.191–2.424) **	38 (11.7)	1.191 (0.837–1.694)	1.142 (0.760–1.716)
**Frontline worker**									
No	236 (5.7)	1	1	451 (10.9)	1	1	408 (9.9)	1	1
Yes	33 (7.3)	1.555 (1.189–2.035) **	1.943 (1.223–3.087) *	73 (16.0)	1.304 (0.970–1.752)	1.986 (1.412–2.793) **	57 (12.5)	1.301 (0.968–1.748)	1.900 (1.317–2.741) **
**Days of quarantine**									
7 days and less	187 (5.7)	1	1	394 (12.0)	1	1	340 (10.4)	1	1.047 (0.837–1.311)
More than 7 days	82 (6.3)	1.116 (0.854–1.459)	1.166 (0.884–1.539)	130 (10.0)	1.226 (0.994–1.511)	1.257 (1.009–1.565) *	125 (9.6)	1.084 (0.874–1.346)	1
**Nationality**									
Non-Malaysian	14 (4.1)	1	1	22 (6.5)	1	1	27 (7.9)	1	1
Malaysian	255 (6.0)	1.491 (0.861–2.584)	1.025 (0.582–1.805)	502 (11.8)	1.943 (1.249–3.023) **	1.432 (0.910–2.254)	438 (10.3)	1.337 (0.891–2.005)	0.952 (0.626–1.448)
**Occupation**									
Not working or retired	31 (5.7)	1	1	58 (10.7)	1	1	54 (9.9)	1	1
Not stated	13 (8.2)	1.481 (0.755–2.903)	1.295 (0.645–2.599)	24 (15.2)	1.498 (0.897–2.501)	1.317 (0.775–2.238)	24 (15.2)	1.622 (0.967–2.721)	1.351 (0.790–2.309)
Student	125 (8.7)	1.573 (1.047–2.364)	1.115 (0.692–1.794) *	235 (16.7)	1.682 (1.239–2.285) **	1.304 (0.908–1.873)	208 (14.8)	1.576 (1.148–2.165) *	1.054 (0.728–1.526)
Working	103 (4.2)	0.718 (0.475–1.085)	0.698 (0.439–1.108)	207 (8.4)	0.764 (0.562–1.039)	0.620 (0.438–0.879) *	179 (7.2)	0.707 (0.515–0.977) *	0.591 (0.412–0.847) **

Note: * = *p* value < 0.05; ** = *p* value < 0.005; CI = confidence interval; OR = odds ratio.

## Data Availability

The datasets generated and/or analysed during the current study are not publicly available due to privacy and third-party restrictions.

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
