# Peer review of "Assessing Mental Health Outcomes in Quarantine Centres: A Cross-Sectional Study during COVID-19 in Malaysia"

_healthcare, 2023, doi:10.3390/healthcare11162339_

Round 1

Reviewer 1 Report

This study aims to assess the mental health-related issues of persons under surveillance and frontline workers in quarantine centers during the second wave of the COVID-19 pandemic in Malaysia.

Examining perspectives from persons under surveillance (PUS) is highly valid. I also find this study's statistical and analytical methods clear and appropriate. This study is beneficial as a resource for developing future countermeasures against the COVID-19 pandemic.

Major Comments

 It is stated that 49 quarantine stations were surveyed during the study period. The number of study subjects, affiliations, and demographics are also clearly stated. I believe that the status of the research subjects is the heart of the study and should be presented more clearly. I would recommend that changes in the number of study subjects throughout the study be indicated using a flowchart diagram. I believe that the flowchart diagrammatically showing the study flow up to the determination of the number of subjects for the analysis will help the reader understand the study better. Please consider this.

In this study, 5454 individuals who provided mental health screening data were included in the analysis. Was there any bias between the group not included in the calculation (the group that did not offer screening data) and the group that had in the analysis? I believe it is essential to indicate this in the paper to determine who is eligible for the study. Please consider this.

Minor Comments

In Figure 1 (L175-176) and Figure 2 (L204-205), the units of each graph are not indicated. I believe clarifying the graphs' units will help readers understand better. Please consider this.

Table 1 (L167) is listed as Table 1 but should be Table 3. Please confirm.

Thank you in advance for your consideration of the above.

Reviewer 2 Report

The manuscript (ID: healthcare-2450466) aimed to assess mental health outcomes and associated factors among persons under surveillance and frontline workers at quarantine centres during the second wave of COVID-19 pandemic in Selangor in Malaysia.    

This cross-sectional study was conducted across 49 quarantine centres in Malaysia, and included 4,577 respondents. The prevalence of stress, anxiety, and depression was found to be 0.9%, 11.4%, and 10.2% respectively.   

But, some issues in this manuscript require major revision (the Methods section):   

  • Lines 14-26:  In the Abstract, the importance of assessing the mental health problems of persons in quarantine centers during the COVID-19 pandemic is indicated, emphasizing that there is a gap in research on this topic. Also, the design of the study, the target population, and the key results are presented.    
  • Lines 17-20: A suggestion for Abstract - Explain whether this paper presents the results of secondary processing of data collected during a cross-sectional study from the Mental Health and Psychosocial Support Services (MHPSS). If so, reformulate this sentence.        
  • Lines 37-38: The words `(citations ... citation)' are probably a reminder to the authors, or not. Correct this. 
  • Lines 30-61: The Introduction section concisely presents the circumstances that represent the background of the investigated problem, with special reference to the activities undertaken in Malaysia in order to manage the negative effects of the COVID-19 pandemic on the mental health of people in quarantine centers. At the end of the Introduction, the clearly defined goals of this paper are presented.    
  • Lines 62-131: The methodology of the work is described in great detail and adequate to the objectives of the research conducted from March to August 2020, with appropriate study design, description of the target population, variables and outcomes, the use of a validated questionnaire (with citation of relevant references), appropriate statistical procedures. Notes:
    • List the inclusion and exclusion criteria of respondents in this study,   
    • Emphasize the method of recruiting respondents in the study,
    • Emphasize that the questionnaires were self-reported.    
  • Line 131: In the section on the methodology of this study, introduce a new subsection in which ethical issues will be explained. The study was conducted during the second wave of the COVID-19 pandemic in Selangor. Was written-informed-voluntary consent to participate in this study obtained at any time during the conduct of this study? If not, explain.
  • Lines 207-216 and Lines 218-221: Move that text to the Introduction section. Begin the Discussion section with the sentence on Lines 216-218.  
  • Lines 222-294: As a whole, this text in the Discussion section is written in a very high-quality way, according to the logical flow, the results of this study are compared with the results of similar research in other countries, appropriate explanations are included that substantiate the presented results, with the citation of appropriate references.     
  • Lines 295-302: The authors introduce a special section `Recommendation for policy makers` in which they listed certain important measures to mitigate negative effects for the mental health of those in quarantine facilities in future epidemics or pandemics.
  • Lines 303-313: In the Limitations section, all the circumstances related to the limitations of this study are discussed in great detail. It is very commendable to see that such questions are presented in an academic manner and honestly in this study.   
  • Lines 314-324: In the Conclusions section, the important results of this study are highlighted correctly. Overall, the findings of this study could be used as a baseline data on how to support the health wellbeing of persons at quarantine centres in the future.       
  • Manuscript in whole: Avoid words such as `we analysed; we considered ...; we discovered; we could …; we used  ...; ...`. Correct it throughout the work.

The quality of English language is appropriate. 

Round 2

Reviewer 2 Report

The authors have satisfactorily and very thoroughly responded to all my questions and made the necessary changes to the manuscript. 

I would like to thank the authors for addressing my comments.  

The quality of English language is appropriate.